# Consumer Response to Cake with Apple Pomace as a Sustainable Source of Fibre

**DOI:** 10.3390/foods10030499

**Published:** 2021-02-26

**Authors:** Ana Curutchet, Julieta Trias, Amparo Tárrega, Patricia Arcia

**Affiliations:** 1Área de Ciencia y Tecnología de Alimentos, Facultad de Ingeniería y Tecnologías UCU, Comandante Braga 2715, CP 11600 Montevideo, Uruguay; ana.curutchet@ucu.edu.uy (A.C.); julieta.trias@correo.ucu.edu.uy (J.T.); 2Instituto de Agroquímica y Tecnología de Alimentos (IATA-CSIC), 46980 Paterna, Valencia, Spain; atarrega@iata.csic.es; 3Latitud Foundation LATU (Laboratorio Tecnológico del Uruguay), Av Italia 6201, 11500 Montevideo, Uruguay

**Keywords:** by-product, cake premixes, acceptability, information

## Abstract

The use of apple pomace flour (APF) as a fibre enrichment strategy was investigated. The aim of this study was to evaluate consumers’ response to intrinsic and extrinsic properties of a bakery premix product when using APF. Apple pomace, a by-product from the juice industry, was dried and ground. APF is high in carbohydrates (47.47%) and fibre (38.48%), and it was used to partially substitute wheat flour and sugar in a cake premix. Acceptability, health, and nutrition questions were evaluated with and without information in terms of regular and fibre-enriched cake. The regular cake score was not affected by information, while the enriched cake’s score increased with information. Three clusters were identified. Cluster 1 (29%) showed high liking scores for regular cake, cluster 2 (31%) for the fibre-enriched cake, and cluster 3 (40%) showed similar liking for both. Consumers described the samples and ideal cake using a check-all-that-apply (CATA) questionnaire. Penalty analyses explained differences in acceptability among consumers. Healthiness, tastiness, and fibre content were the main reasons to buy the enriched cake for cluster 2; taste for consumers in cluster 1; and healthiness and taste for consumers in cluster 3. APF as a functional ingredient may be a consumers’ choice as a sustainable use of apple pomace.

## 1. Introduction

Apples are one of the most popular fruits in the world and are widely grown in regions of all temperatures. Worldwide, apple processing generates a huge volume of waste, considering the annual processed tonnage up to 11 million tons (Mt) [1]. Apple pomace is the pressed residue obtained after processing apples into juice, cider, distilled spirit, and vinegar. The solid waste consists of the core, peel, seed, pulp, and kernel of the fruit and represents 20–35% of the fresh weight of the apples [2]. According to the tonnage processed, 3.3 million tons (Mt) of waste are generated every year.

This by-product is usually treated in traditional ways, such as landfilling, incineration, composting, and low-quality animal feed, causing serious environmental problems and losses for the industries due to the waste treatment and transportation costs for dumping into landfills [3,4]. Thus, it is of vital importance to reuse industrial by-products to improve the process economics and its sustainability.

Apple pomace is an important source of dietary fibre, since nearly 40% of its dry weight is fibre [5]. According to [6], dietary fibre is a remnant of the edible part of plant; it is analogous to carbohydrates that are resistant to digestion and absorption in the human small intestine and undergo complete or partial fermentation in the large intestine. Dietary fibre has many health benefits and could help in the treatment of obesity, atherosclerosis, coronary heart diseases, large intestine cancer, and diabetes by decreasing blood glucose level. It can bind hydrochloric acid, metal ions, and cholesterol in the stomach and can also stimulate growth of probiotic microflora in the intestines [7]. An adequate intake for total dietary fibre is set at 28 and 25 g per day for young men and women, respectively [8].

Due to its relative low cost and potential nutritional value, apple pomace has been considered as an attractive functional ingredient to formulate human food. In fact, Alongui et al. [5] developed short dough biscuits with partial substitution of wheat flour by apple pomace flour which allows its glycemic index to be reduced. Masoodi et al. [9] studied batter characteristics of cakes enriched with apple pomace, and Rupasinghe et al. [10] evaluated the impact of baking in fibre and antioxidant capacity of muffins incorporated with different levels of apple skin powder.

However, the success in the market of a product including a new ingredient requires consumers’ acceptance. Intrinsic product properties such as sensory properties are considered determining factors of acceptance and choice of a product, but the extrinsic properties—the information on the label such as brand, nutritional facts, and claims—can also play an important role in consumer buying behaviour [11,12,13,14,15]. Previous studies have shown how characteristics related to the label can influence, positively or negatively, consumer expectation and hedonic evaluation of food products [16,17,18,19,20,21]. Concerning information about fibre content, its impact has been investigated for baguettes and bread rolls by [20,21], respectively. The first study noticed that a “source of fibre” label in French baguettes had a positive effect on willingness to pay, while the second study observed that some consumers (clusters) even preferred bread with higher content of fibre. In the same way, the authors of [22] studied the effect of information about fibre in three types of muffins (plain, wholemeal, and enriched with resistant starch), noticing that the score in texture and overall acceptance were increased in muffins when nutritional information was provided. Many studies between 2017 and 2019 used apple pomace in bakery products. It the effect of apple pomace in nutritional profile, food colour, and texture properties has been studied. There are even some acceptability studies in cookies and gluten-free crackers [23]. However, it has not been studied in depth how consumers perceive apple fibre.

As multiple elements could affect preference and liking, it is necessary to provide a view that integrates as many factors as possible in product development, especially in terms of functional food. Thus, the aim of the present work was to evaluate consumers response to intrinsic and extrinsic properties of a bakery premix product when using apple pomace for fibre enrichment.

## 2. Materials and Methods

### 2.1. Apple Pomace Flour Obtention

Apple pomace was obtained as a by-product from juice production. The apples were Royal Gala, Early Red OneRed Chiff, Dana Red, and Granny Smith; were harvested and processed in Uruguay; and were provided by MIS OLIVOS S.A (Montevideo, Uruguay). The apple pomace flour (APF) was obtained by drying apple pomace in a convection oven at 70 °C for 8 h and grounding in a laboratory mill, using the fraction that passed through a 1 mm sieve. APF was stored at −18 °C in a cold chamber.

### 2.2. Apple Pomace Proximate Composition

Proximate analyses were performed on the apple pomace flour (APF). Protein was determined by the copper catalyst Kjeldahl method proposed by the Association of Official Analytical Collaboration (AOAC, 984.13) and total dietary fibre (TDF) was determined using the AOAC enzymatic gravimetric method (985.29) [24]. Fat content was estimated using the Soxhlet procedure described by [25]. Moisture content was determined by gravimetric analysis in a convection oven at 105 °C until constant weight. Ash was determined in a muffle furnace following the work of [26]. Total carbohydrate content was obtained by difference between the total weight and the sum of grams of protein, lipids, dietary fibre, moisture, and ash content in 100 g of sample.

### 2.3. Premix Formulations and Cake Preparation

Apple pomace flour was used to prepare a pre-mix of fibre-enriched cake. Developed premix formulation included wheat flour, sugar, baking powder, skim milk powder, and vanilla flavour (Table 1). The amount of apple pomace (23.18 g per 100 g) was adjusted to reach a fibre content of 6.3 g/100 g, with the aim of labelling the product with the claim “source of fibre”, which corresponds to 2.5 g of dietary fibre per serving, according to the Uruguayan legislation [27].

A regular cake was prepared using a commercial pre-mix brand (sugar, wheat flour, corn starch, whey protein, glucose syrup, baking powder, vanilla flavour, carboxymethylcellulose, content non-declared) with the highest acceptably among cake premixes in the Uruguayan market [28].

Cakes were prepared with the premix and adding vegetable oil, eggs, and water, following the proportions established in Table 1. Nutrition facts for both cakes were calculated (Table 2). Ingredients were obtained from the local market.

### 2.4. Cake Labels

The packaging of both cakes was designed using the label of the commercial cake premix as a reference, one that is a brand leader in the Uruguayan market. The original packaging was adjusted in such a way that the only difference between both products were nutritional facts (Table 2), claims, and cake aesthetic. The claim “source of fibre” was included as well as the “with apple fibre” reference in the fibre-enriched cake. The labels of both cakes are shown in Figure 1.

### 2.5. Consumer Evaluation Sessions

Two consumer sessions were developed with the fibre-enriched cake and regular cake, namely, blind and informed tests. Both evaluation samples were served to consumers on plastic plates, and were coded using random three-digit numbers. Water was available for consumers.

#### 2.5.1. Blind Test

In these acceptability tests, consumers evaluated the product without any information. The session was carried out by 104 consumers (58% men and 42% women) whose ages ranged between 17 and 60 years old. Overall acceptability was evaluated with a nine-point hedonic scale, ranging from 1 (“I dislike extremely”) to 9 (“I like extremely”).

#### 2.5.2. Informed Test

A second evaluation including product packages was performed to evaluate the impact of information on the response of consumers (acceptability and purchase intention).

A total of 102 consumers, whose ages ranged between 18 and 70 years old, evaluated the cakes with the label information presented as an image on a computer screen. For each sample, consumer was asked to observe the label, taste the sample, and then rate acceptability on a nine-point scale ranging from 1 (“I dislike extremely”) to 9 (“I like extremely”) and purchase intention on a 5-point scale from “I would definitely buy it” to “I would definitely not buy it”. After that, a check-all-that-apply (CATA) questionnaire was used to describe the sample. The list of attributes in the CATA questionnaire was determined in a preliminary essay with 10 consumers and included sensory and non-sensory characteristics (homemade, flattened, tasty, aftertaste, high in calories, dry, moist, distasteful, gritty, soft, intense flavour, tasteless, fruity flavour, healthy, sweet, spongy, easy to chew, bitter, fibrous, odd flavour, hard to chew).

According to the response to the purchase intention, we asked consumers to indicate the reasons for either buying or not buying the cake by selecting the appropriate ones from a list previously generated in a preliminary essay with 10 consumers (Table 3).

Consumers were also asked their frequency of consumption of cakes on a four-point scale from 1 (“never”) to 4 (“once a week or more”), After that, participants completed the General Health Interest questionnaire proposed by Roininen et al. [29] using the Spanish version reported by Villegas et al. [30]. Finally, respondents completed some demographic questions and noted their gender and age. Data collection was conducted using Compusense Cloud (Compusense Inc., Guelph, ON, Canada).

### 2.6. Data Analysis

All the results are expressed as mean ± SD. Student’s *t*-test was performed to compare overall acceptability and purchase intention of samples. To identify groups of consumers with different patterns, we applied hierarchical cluster analysis considering Euclidean distances and Ward’s aggregation method. Composition of each cluster according to consumer gender, age, and interest in a healthy diet were compared using the chi-squared test. Significant differences among proportions were determined using the Marascuilo procedure [31]. A non-parametric Mann–Whitney test was performed to compare acceptability between samples into clusters.

The frequency of mention of each CATA term was determined. Cochran’s Q test was performed on the binary CATA data to determine significant differences between samples for each attribute (*p* ≤ 0.05). To determine the impact of each attribute on liking, we used penalty-lift analysis. For this, liking is averaged across all observations (consumers) in which the attribute was used to describe the product, and across these observations for which it was not used. The difference between these two mean values provides an estimate of the average change in acceptability due to the presence of the attribute. The same procedure was replicated in each sample (regular and fibre enriched cake).

Analyses were performed using XLSTAT 2020.3.1 (Addinsoft 2020, New York, NY, USA).

## 3. Results and Discussion

### 3.1. Apple Pomace Flour Characteristics as an Ingredient

Proximate composition of the APF, including the contents for protein, fat, ash, dietary fibre, and carbohydrate content is shown in Table 4. As notable features, APF includes high contents of dietary fibre and carbohydrates from fruit and low-fat content. According to this, APF is an interesting ingredient to substitute part of the wheat flour in sweet bakeries. It allows incorporating fibre to reduce sucrose content. In the same way, the authors of [32] used APF as an ingredient with high levels of polyphenols and also phytate-free dietary fibre to develop gluten-free cakes. The authors of [23] developed a systematic review of apple pomace as a food fortification ingredient. They concluded from analysing data from 2007 to 2019 that APF has a beneficial nutritional profile and is an ecological issue.

For the above-mentioned reasons, we decided to use apple pomace as a functional ingredient in the development of a bakery pre-mixes, evaluating its acceptability and the effect of information on consumers’ response in comparison with commercial pre-mixes.

### 3.2. Acceptability of Cakes in Blind and Informed Conditions

Means of acceptability for the regular and fibre-enriched cakes in blind and informed conditions are shown in Table 5. In the blind condition, when consumers only tasted samples, mean of liking scores for the fibre-enriched cake was significantly lower than for the regular cake. This difference in liking may have been due not only to the presence of fibre but also to the fact that regular cake was higher in fat and sugar content, which are usually related to higher overall acceptability [33]. The amount of added sugar was kept lower than in regular cake because it was expected that there would be a contribution of apple pomace to sweetness due to its high sugar content and because we were trying to develop a healthier product.

By comparing blind and informed conditions, we found that label presence significantly increased consumers’ liking for the fibre-enriched cake, while for the regular cake, label presence only caused a slight increase in acceptability that was not significant. This indicates claims that “fibre source” and “apple fibre” increase acceptability for cake. According to [34], product information may influence consumer’s expectation and choice. The authors found a positive effect of nutritional information on consumers’ acceptability of pasta produced with the addition of wheat bran, and similar results were obtained by [22], who observed that overall acceptability increased when information was provided for a wholemeal muffin.

Despite the increase observed for fibre-enriched cake, we found a lower liking score than the regular cake. However, the percentage of consumers buying fibre-enriched cake (53%) was close to that observed for regular cake (63%).

### 3.3. Individual Variability in Consumers’ Response to Apple Fibre-Enriched Cake

Cluster analysis was applied to identify segments of consumers with different patterns of preference with respect the regular and fibre-enriched cakes (evaluated in the presence of the label). Three clusters were identified (Table 6). Consumers in cluster 1 (29%) showed high liking scores for regular cake and low liking scores for the fibre-enriched cake. Only 13% of them would buy the fibre-enriched cake and 56% would buy the regular cake. Cluster 2 (31%) preferred the fibre-enriched cake than the regular cake. A total of 81% of these consumers would buy the fibre-enriched cake and 41% would buy the regular cake. Finally, cluster 3 (40%), despite showing high liking scores for both cakes, liked the regular cake more. Thus, while a high percentage of consumers would buy the regular cake, many of them would also buy the fibre-enriched cake (59%).

Previous studies have shown that inter-individual differences are especially relevant in the response of consumers to fibre-related information and claims in products such as bread, muffins, yogurt, and cakes [22,35,36]. In the same way, the authors of [35,36] found culture-related differences in the impact of fibre information on liking and/or sensory properties of food, and the authors of [22] found that fibre-related information was more effective in increasing liking of muffins for health-conscious consumers. In the present study, clusters did not significantly differ in terms of gender (*p* = 0.360) and health interest (*p* = 0.861), but they differed in terms of age (*p* = 0.019). Cluster 1 and 3 were mainly composed of people younger than 35 (66% and 61%), and cluster 2 showed more people older than 35 (58%). It seems that in the case of fibre, older people show more willingness to accept a new product than young people. Similarly, the authors of [20] found a significant age effect on the willingness to pay for high-fibre bread. According to these authors, young consumers are more influenced than the older consumers by the hedonic value of the bread more than health concerns and they are not willing to sacrifice pleasure for health.

### 3.4. Consumer Drivers of Liking and Reasons to Purchase

Consumers also described (CATA questionnaire) the cakes after tasting them and observing the labels. For 15 of the 21 terms in the list, significant differences were observed in terms of the frequency of mention. Figure 2 shows the frequency of mention of the two cakes compared to the ideal cake, which was described by most of the participants as being homemade, tasty, moist, sweet, soft, spongy, and healthy. The fibre-enriched cake showed a higher frequency for some of these terms, such as being homemade and healthy, while the regular cake showed higher frequency in terms of being sweet and soft. Both cakes were described as tasty and easy to chew. Among the terms not cited for the ideal cake, dry and tasteless were used similarly for both cakes by some consumers (15–18%), and fibrous and sandy were used by some consumers only for the fibre-enriched cake.

To understand the variability in consumers response, we applied a penalty lift analysis to determine the attributes that explain the differences in consumers liking scores for the same product (Figure 3). For the fibre-enriched cake, the acceptability was higher for consumers that described it as tasty, homemade, and healthy, and lower for consumers that found it tasteless or dry. For the regular cake, the variation in liking scores among consumers was mainly associated with positive terms; the consumers who found the cake tasty, homemade, easy to chew, soft, and spongy gave it a higher liking score, while those who found the cake dry gave it a lower liking score (Figure 3).

Figure 3 shows global results in which most attributes had a positive connotation in cakes. As expected, the attributes tasty, soft, homemade, easy to chew, and moist were found to be the most important drivers of liking, increase liking by around 1 point on the nine-point scale when they were present. On the other hand, the worst impact in acceptability was given by the attributes flattened, distasteful, and bitter.

In sponge cakes enriched with fibre from different sources, the authors of [37] reported a similar positive impact of easy to chew, spongy, soft, and sweet on liking. Negative impact of dry and tasteless sensations on liking were also found by these authors, but they also found a negative impact of soft, odd flavour, and fruity flavour (especially for blackcurrant fibre cake) that was not found in this work for the cake enriched with apple fibre. This seems to be an advantage for apple fibre because, as reported by [38], certain flavours provided by the addition of fruit fibre can make the product too different from original to be acceptable. It is important to note that while in this work, homemade and healthy were relevant attributes with positive impact for the fibre-enriched cake, they were not found to be relevant in the study of [37]; one possible explanation for this difference is that in this study, cakes were evaluated without any product information.

As expected, the reasons that consumers linked to their buying decisions were in agreement with the drivers of linking. Among consumers buying the fibre-enriched cake (53%), the main reasons were that it is healthy (27%) and tasty (35%) (Figure 4). As observed in Figure 4, being healthy and tasty were relevant reasons for consumers in cluster 2, but for consumers in cluster 3, being tasty was more relevant. Among those buying regular cake (67% of consumers), being tasty was the reason indicated for almost all of them (60% of consumers). For both fibre-enriched and regular cake, the main reasons for not buying the cake were that it was not tasty (23% and 11%, respectively), but also to prevent weight gain (8%). Additionally, some consumers did not buy the fibre-enriched cake because it was not the same as what they were used to, and some consumers did not buy regular cake because they consider it unhealthy.

In the same way, the authors of [39] studied fibre-enriched breads and found two groups of consumers with different consumption choices: hedonistic consumers whose motives were driven mainly by “mood” and “price”, and traditional consumers driven by “natural content”, “familiarity”, and “health concern” claims.

Regarding reasons to buy, the authors of [40] found that price, sensory appeal, convenience, and health were strong motives contributing to food choices. These authors also reported that young adults who value the food choice motives of weight control and natural content are likely to have a very positive attitude towards functional food. As the authors of [41] proposed, young adults experience many barriers to living a healthy lifestyle—high stress, poor sleep, and challenges in time balance—which may increase the need for convenience in food choices.

APF was used as a fibre enrichment strategy, and as Figure 1 shows, consumers were able to notice that the product contained apple fibre, but they did not have information about the apple fibre origin. Consumers did not have explicit information about fibre-enriched cake sustainability. Kaczorowska et al. (2019) [42] found that sustainability labels influence consumer buying behaviour in spite of the fact that consumers perceive the benefits of buying them differently. In these ways, it should be explored whether sustainability should be a valuable reason to rise up the lower acceptability of fibre-enriched cake.

## 4. Conclusions

Due to its high fibre content, apple pomace flour, obtained by mild processing conditions, shows a high potential as an ingredient to produce bakery goods with healthier formulation. The cake enriched with apple pomace flour was less liked by consumers than regular cake, but liking was shown to increase when the product packaging contained “fibre source” and “apple fibre” claims and nutritional facts. High individual variability in consumers’ response to fibre-enriched and regular cakes was observed. Clustering consumers’ response allowed us to identify a group (30% of participants) that preferred fibre-enriched cake over regular cake because they considered it healthy, tasty, and as if it were homemade, as well as providing the fibre they needed in their diet. Among consumers preferring regular cake, some of them were also interested in the fibre-enriched cake, mainly because they liked it as well. Despite young people being usually more willing to accept new products, in this case, the consumers who were not interested in buying fibre-enriched cake were mainly young people (<35 years old) and the main reason was that they did not like the sample. Improving sensory characteristics such as increasing sweetness, moisture, sponginess, and softness sensation may be a way to increase the success of the product.

Apple pomace flour may have a place among consumers. This may be a promissory use of apple pomace. The large amount of waste generated by this by-product makes finding alternative uses a priority.

## Figures and Tables

**Figure 1 foods-10-00499-f001:**
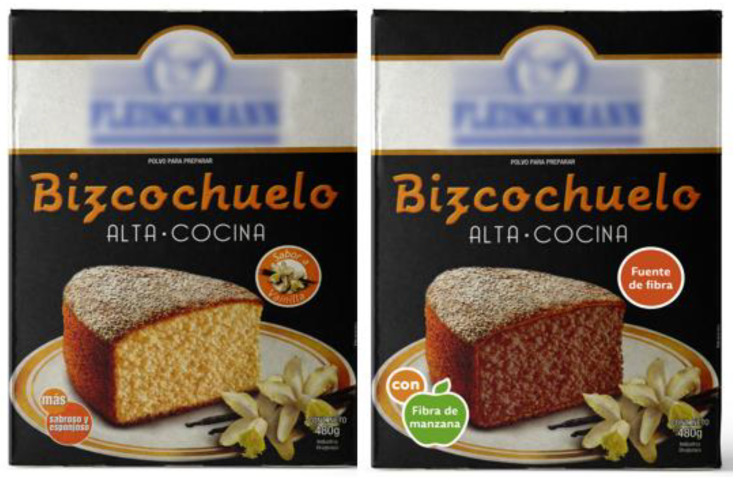
Product packaging design.

**Figure 2 foods-10-00499-f002:**
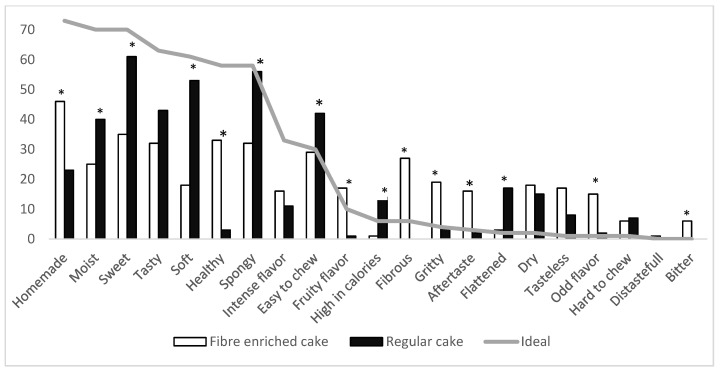
Frequency of check-all-that-apply (CATA) terms for the fibre-enriched, regular, and ideal cakes. CATA terms with an asterisk have significant differences between regular cake and fibre-enriched cake; *p* < 0.05.

**Figure 3 foods-10-00499-f003:**
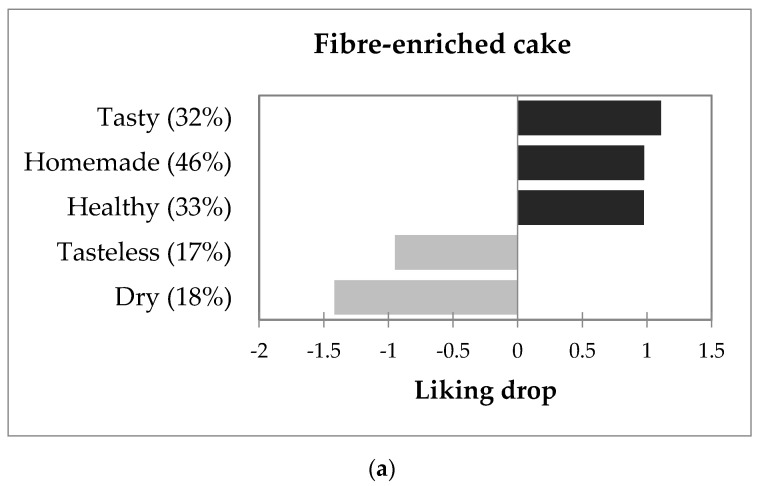
Impact of the attributes on the acceptability of the regular (**b**) and fibre-enriched cakes (**a**) according to penalty lift analysis. For each attribute, the percentage of consumers that selected it for describing the product is indicated in brackets.

**Figure 4 foods-10-00499-f004:**
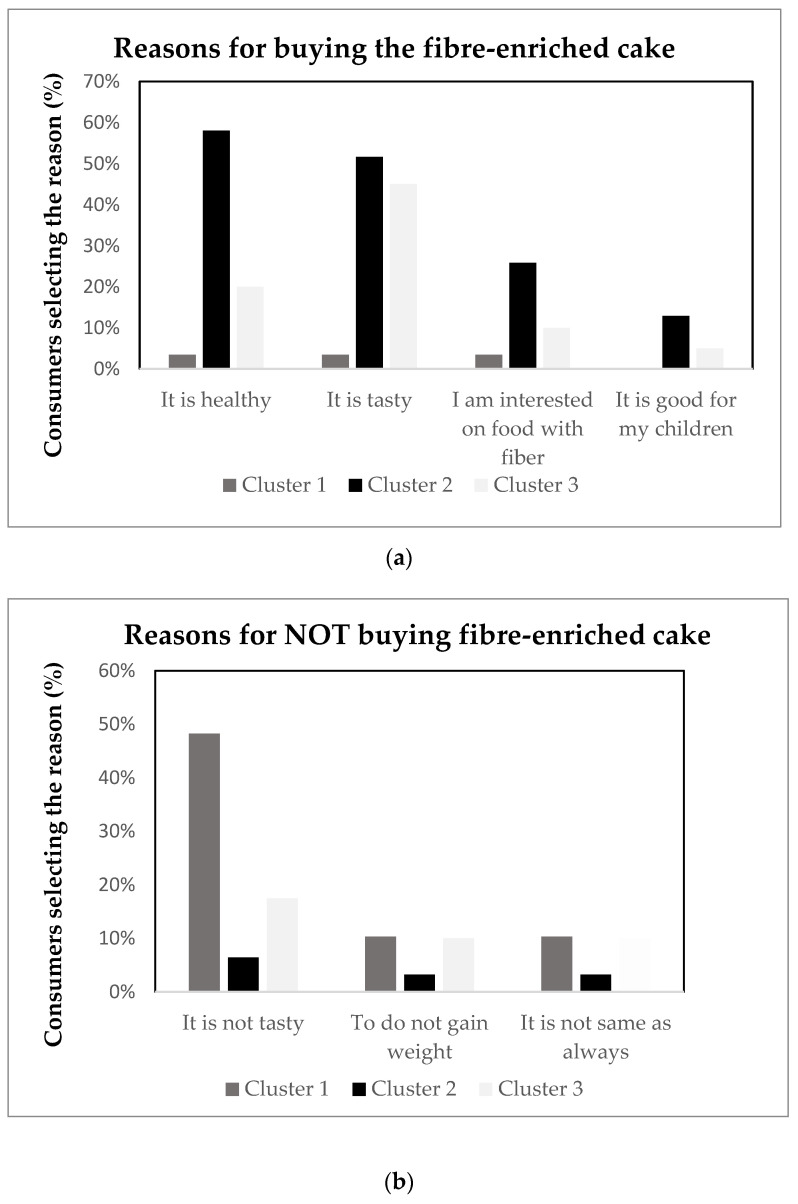
Reasons to buy (**a**) or not buy (**b**) cakes evaluated with information between three clusters. Reasons selected for more than 10% of consumers are included.

**Table 1 foods-10-00499-t001:** Formulated fibre-enriched cake premix composition.

Ingredients	Amount in Premix (%)	Amount in the Cake (%)
Wheat flour	35.67	20.55
Apple pomace flour	23.18	13.36
Sugar	31.21	17.98
Baking powder	4.46	2.57
Skim milk powder	3.57	2.05
Vanilla flavor	1.92	1.10
Vegetable oil	-	7.71
Eggs	-	12.84
Water	-	21.83

**Table 2 foods-10-00499-t002:** Nutrition facts for fibre enriched and regular cake expressed as content in 100 g of sample.

Ingredients	Fibre Enriched Cake	Regular Cake
Energy	275 kcal = 1149 kJ	332 kcal = 1388 kJ
Carbohydrate	43 g	52 g
Protein	4.8 g	11 g
Total fat	9.5	8.7 g
from which saturated	1.1 g	3.2 g
Dietary fibre	6.3 g	1.0 g
Sodium	280 mg	448 g

**Table 3 foods-10-00499-t003:** Question of purchase intention: reasons for buying or not buying the cake.

**Why would you buy it?**
1. Because it is healthy
2. Because it is tasty
3. Because it is the one I always buy
4. Because I don’t need fibre in my diet
5. Because it is low in calories
6. To avoid weight gain
7. Because it is good for my kids
8. Because I want to include fibre in my diet
9. Because my kids would like it.
**Why wouldn’t you buy it?**
1. Because it is not healthy
2. Because it is distasteful
3. Because it is not the one I always buy
4. Because I need fibre in my diet
5. Because it is calorific
6. To prevent weight gain
7. Because it is bad for my kids
8. Because I don’t want to include fibre in my diet
9. Because my kids wouldn’t like it.

**Table 4 foods-10-00499-t004:** Proximate composition of the apple pomace g/100 g.

Proteins	3.04 ± 0.18
Lipids	3.08 ± 0.08
Ash	2.07 ± 0.10
Dietary fibre	38.48 ± 0.20
Carbohydrates	47.47 ^1^

^1^ Carbohydrate content was obtained by difference.

**Table 5 foods-10-00499-t005:** Acceptability of cakes under blind and informed conditions and purchase intention.

Product	Blind Condition	Informed Condition	Intention to Purchase
Fibre-enriched cake	5.34 ± 1.80 ^a,A^	6.55 ± 1.58 ^a,B^	3.42 ± 1.02 ^a^
Regular cake	6.70 ± 1.74 ^b,A^	7.06 ± 1.50 ^b,A^	3.69 ± 0.99 ^b^

For the same column, scores not sharing letters are significantly different (*p* ≤ 0.05) according to Student’s *t*-test. For the same row, scores not sharing a capital letter row are significantly different (*p* ≤ 0.05) according to Student’s *t*-test.

**Table 6 foods-10-00499-t006:** Cluster composition according cake acceptability, cakes’ frequency consumption and consumer´s age distribution.

Product	Cluster 1 (29%)	Cluster 2 (31%)	Cluster 3 (40%)
Apple fibre cake	4.5 ^a^	7.4 ^a^	7.3 ^a^
Regular cake	6.5 ^b^	5.9 ^b^	8.3 ^b^
Age (>35 years) *	35 ^A^	58 ^B^	39 ^A,B^

Mean acceptability scores of consumers (Clusters 1–3). Scores not sharing letters within each cluster were significantly different (*p* ≤ 0.05) according to Mann-Whitney test. * Scores not sharing capital letters are significantly different (*p* ≤ 0.05) according to the chi-square test. Significant differences among proportions were determined using Marascuilo procedure.

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
