# Peer review of "Consumer Response to Cake with Apple Pomace as a Sustainable Source of Fibre"

_foods, 2021, doi:10.3390/foods10030499_

Round 1

Reviewer 1 Report

After reviewing the manuscript “Consumers response to cake with apple pomace as source of fibre”, the manuscript presents a well structure and correct presentation of results. English language and style a little require editing.

The work is very interesting. A very important topic for waste management as an alternative to a food product.

This are my comments on the presents manuscript:

How long was the pomace flour kept at a negative temperature? So, after what time were the oil stages completed?

Is it correct to say "response to intrinsic and extrinsic properties of a bakery- premix product when using apple pomace for fiber enrichment." Is the organoleptic characteristics or sensory analysis of the product better?

It is very unclear: "Protein and Total dietary fiber (TDF) were determined using AOAC methods [24], 984.13, 985.29 respectively. Fat was estimated following the [25] procedure. ...." the first sentence and the second piece are unclear and you don't know what the numbers are about. In addition, the procedures should be briefly described.

It was necessary to better describe the results and there is no greater discussion with the results given by other authors. Maybe try to combine the results or represent it in a different graph (not a bar)?

A mistake crept into the conclusions, and more the conclusions.

This section is not mandatory but can be added to the manuscript if the discussion is unusually long or complex. - please delete.

It is worth to write in points

Was it reported that this flour could also be a source of fruit sugar?

Author Response

Thank you very much for the positive feedback and for taking time to review the paper carefully. Here are our reply to all your comments:

How long was the pomace flour kept at a negative temperature? So, after what time were the oil stages completed?

  • Apple Pomace Flour was kept at -18 °C for 3 weeks while it was used for preliminary analysis and cake premixes production. Proximate composition (including fat content determination) was determined the first week.According to reviewer suggestion, this information has been included in the manuscript.

Is it correct to say "response to intrinsic and extrinsic properties of a bakery- premix product when using apple pomace for fiber enrichment." Is the organoleptic characteristics or sensory analysis of the product better?

  • Authors have changed the text:

    "Thus, the aim of the present work was to evaluate consumer’s response to fibre enrichment with apple pomace in a bakery- premix product." Line 96.

    It is very unclear: "Protein and Total dietary fiber (TDF) were determined using AOAC methods [24], 984.13, 985.29 respectively. Fat was estimated following the [25] procedure. ...." the first sentence and the second piece are unclear and you don't know what the numbers are about. In addition, the procedures should be briefly described.
  • Following reviewer suggestion, the sentence was corrected and the methods were briefly explained. Line 108-113.

    It was necessary to better describe the results and there is no greater discussion with the results given by other authors. Maybe try to combine the results or represent it in a different graph (not a bar)?

  • Conclusion was kept in a separated section since the discussion is quite long.

Was it reported that this flour could also be a source of fruit sugar?

  • Despite the fact that it could be used as a source of fruit sugar, none of the papers revised declare use as such. all findings show it was used as source of fiber. Note that given that we noted that several papers quantified large amounts of fruit sugar, we decided to change the formulation and reduce the added sugar. [2],[4],[5].

Reviewer 2 Report

Review of the manuscript foods 1108036:

Consumers response to cake with apple pomace as source of fibre

Overall

The sustainability and use of waste streams should be highlighted also in the discussion. To make the aim clearer some of the text should be re-arranged.  Language needs to improve.

Title

The title is informative and clear. Could be improved by adding the word sustainable, eg “Consumers response to cake with apple pomace as a sustainable source of fibre”. However, that is up to the authors to decide.

Abstract

Abstract is well written and gives useful information. However, the aim is not stated and the word CATA question should be changed into CATA questionnaire. Please add aim and change the word.

Introduction

Interesting and well written, the text leads toward the aim.

Material and Methods

2.2. Apple pomace proximate composition

Since the aim of the study is to anlayse consumer response, why this section? Suggest to take what is in results and add here as an information of the nutritional content. In the table the analysis method could be mentioned. (Line 90: Define AOAC)

2.5.1 Blind test

Add a reference to the hedonic test.

2.5.2 Informed test

Define CATA when mentioning the abbreviation for the first time.

Change question into questionnaire.

References 29 and 30 should be mentioned by author names.

2.6. Data analysis

Change into Students’ t-test. Why not ANOVA?

Results and Discussion

Overall in this section a discussion on the sustainability and use of waste streams is missing. Please add a couple of sentences of that.

3.1. Apple pomace flour characteristics as an ingredient

The content of this section should be moved to material and method, since analysis of content is not included in the aim.

Conclusions

The conclusion makes sense and should be included in the final version. Please, don’t forget to delete the first sentence!

Author Response

Thank you very much for the positive feedback and for taking time to review the paper carefully. Here are our reply to all your comments:

  • Title was changed to the suggested by the reviewer.
  • Abstract: Aim was explained and included in abstract, CATA question was replaced by CATA questionnaire. All abstract should be reorganized to reach 200 words.
  • Materials and methods: Given the relevance of the apple pomace proximate composition for the overall results, it is of utmost importance that we bring it to the discussion. Apple pomace was used as a source of fiber and flavor to partially replace wheat flour and sugar in a bakery-premix. Though fiber and sugar are relevant macronutrients found in juice industries´ by-products, its content may differ among the different apple varieties and juice technology used. Taking this into consideration, we believe it is important to develop premixes taking into account proximate composition of the apple pomace available in our market and not the composition as detailed in the literature.
  • Materials and methods: AOAC acronym was defined.
  • Materials and methods: question was replaced by questionnaire.
  • Materials and methods: acronym CATA was defined in the abstract.
  • Materials and methods: References 29 and 30 are mentioned by author names.
  • Materials and methods: Blind test was better explained, line 177.
  • Data analysis: As participants were not the same in the two evaluation conditions, authors prefer compare means using t-tests, paired t-test for the same evaluation condition and independent t test for comparing  the two conditions.This information has been now included in the manuscript. Paired samples t-tests were used to compare acceptability means between the two samples in each evaluation condition. Independent t-test were used to compare the acceptability means of two sample, between blind and informed evaluation.
  • Results and discussion: Overall in this section a discussion on the sustainability and use of waste streams is missing. Please add a couple of sentences of that. New paragraph was added line 354-360.
  • Results and discussion: We prefer to maintain Apple pomace flour characteristics as an ingredient in results because we think is important to visualize the by-product characteristics.
  • Conclusions: First sentence, that came from the template was deleted.

Reviewer 3 Report

I would suggest the following:

  1. Line 34: there is no reference.
  2. Line 39: there is no reference.
  3. Write the producer of pre-mix brand.
  4. How consumers were educated for the sensory evaluation?
  5. Line 298: erase it.

The findings are interesting and well desribed.

Author Response

Thank you very much for the positive feedback and for taking time to review the paper carefully. Here are our reply to all your comments:

  • The sentence reference in lines 33 and 34 corresponds to the same reference of the next sentence (lines 34 and 35). Both sentences correspond to reference number 1.
  • The sentence in line 39 has no reference. The amount to which mention is made was calculated by the authors according to the amounts and percentages previously mentioned.
  • Pre-mix used in this work corresponds to a leader brand in the Uruguayan market, authors consider for the work is not valuable to inform the brand.
  • Consumers were instructed to perform each sensory evaluation. Instructions were given in writing.
  • Authors consider that sentence in line 298 is needed because it mentions attributes that worst impact in acceptability.

Round 2

Reviewer 1 Report

After reviewing the manuscript “Consumers response to cake with apple pomace as source of fibre”, the manuscript presents a well structure and correct presentation of results.

Thanks for posting the corrections. I got quite comprehensive answers. It is always possible to expand the issue, but due to the repeatability of the material, there may be problems here.

Author Response

Thank you very much for your positive feedback.

Reviewer 2 Report

Thank you for revision. The manuscript hs now improved and should be published.

Author Response

(The authors gave the same response as above.)
